# The New Synthetic Serum-Free Medium OptiPASS Promotes High Proliferation and Drug Efficacy Prediction on Spheroids from MDA-MB-231 and SUM1315 Triple-Negative Breast Cancer Cell Lines

**DOI:** 10.3390/jcm8030397

**Published:** 2019-03-21

**Authors:** Clémence Dubois, Pierre Daumar, Corinne Aubel, Jean Gauthier, Bernard Vidalinc, Emmanuelle Mounetou, Frédérique Penault-Llorca, Mahchid Bamdad

**Affiliations:** 1Institut Universitaire de Technologie, Université Clermont Auvergne, INSERM, U1240, Imagerie Moléculaire et Stratégies Théranostiques, F-63000 Clermont Ferrand, France; clemence.dubois@uca.fr (C.D.); pierre.daumar@uca.fr (P.D.); emmanuelle.mounetou@uca.fr (E.M.); 2Centre Jean Perrin, Université Clermont Auvergne, INSERM, U1240, Imagerie Moléculaire et Stratégies Théranostiques, F-63000 Clermont Ferrand, France; frederique.penault-llorca@clermont.unicancer.fr; 3Faculté de médecine, Université Clermont Auvergne, INSERM, U1240, Imagerie Moléculaire et Stratégies Théranostiques, F-63000 Clermont Ferrand, France; corinne.aubel@uca.fr; 4Biomarqueurs S.A.S, F-63200 Riom, France; gauthierfifa@orange.fr (J.G.); bernard.vidalinc@orange.fr (B.V.)

**Keywords:** synthetic medium OptiPASS, triple negative breast cancer, MDA-MB-231, SUM1315, spheroid development, drug screening

## Abstract

Triple-negative breast cancers are particularly aggressive. In vitro cultures are one of the major pathways for developing anticancer strategies. The effectiveness and reproducibility of the drug screenings depend largely on the homogeneity of culture media. In order to optimize the predictive responses of triple-negative breast cancer 3D cell culture models, these works were focused on the development of SUM1315 and MDA-MB-231 cell lines in OptiPASS medium, a new serum-free formulation (BIOPASS). In monolayer cell culture, OptiPASS medium was more suitable for MDA-MB-231 than SUM1315 cell line but maintained cell phenotype and allowed sufficient proliferation. For spheroids produced in OptiPASS, the size monitoring showed a 1.3 and 1.5-fold increase for MDA-MB-231 and SUM1315 cell lines, respectively and viability/mortality profiles were maintained. Spheroids drug sensitivity thresholds were also improved allowing quicker high throughput drug screenings. These results showed the suitability of OptiPASS for 2D and 3D cell cultures of these two triple-negative breast cancer cell lines, with reproducibility of spheroid formation superior to 98%. This opens the way to the common use of this synthetic medium in future preclinical breast cancer research studies.

## 1. Introduction

Triple-negative breast cancers (TNBC) are particularly aggressive and of poor prognosis [1,2]. It is a heterogeneous and complex pathology, characterized by the non-expression of estrogen receptors (ER) and progesterone receptors (PR), and the absence of epidermal growth factor receptor-2 (HER2) over-expression or ERBB-2 amplification [3,4]. Recently, five molecular subtypes of TNBC tumors have been described by Lehmann [5] which are (i) basal-1, (ii) basal-2, (iii) mesenchymal-like, (iv) androgen-receptor and (v) immunomodulatory. Currently, therapies specifically targeting the characteristics of these tumors are under development in preclinical and clinical studies [6,7]. In order to model the effectiveness of these targeted therapies, laboratories performing preclinical studies use in vitro cell culture techniques. This methodology allows the reproduction of cell physiological conditions. As a result, it must respect certain parameters essential to cell survival, such as pH, temperature, osmolarity, and oxygen supply. In order to control these parameters, specific culture media have, therefore, been developed, allowing cell survival and proliferation [8]. The culture medium is one of the key factors of in vitro modeling. Thus, there are different categories of culture media, with common characteristics and variants, depending on the type of cells studied [9]. The majority of the cell culture media are supplemented with fetal calf serum (FCS), which is a supplement of animal origin. It is required for culture of mammalian cells as it provides multiple elements essential to initial adhesion, growth, metabolism, and proliferation of cells in the culture [8]. Its composition is complex because of its biological origin, but its main constituents are growth factors, such as insulin, transport proteins, such as albumin, attachment factors promoting cell adhesion, such as fibronectin, and amino acids. It also contains vitamins, such as vitamin B, some essential minerals, lipids, fatty acids and protease inhibitors, which possess an action against proteolytic enzymes, such as trypsin. FCS also acts as a buffer to maintain pH and protects the cells from shocks, thanks to its viscosity [10]. Despite its many advantages, the use of the FCS presents several drawbacks. Indeed, like any compound of biological origin, its general composition is not controllable, resulting in variability between batches. This variability requires the systematic control of batches’ quality in terms of cell growth performance and the absence of contaminants (mycoplasma, viruses, etc.) or growth inhibitors. Furthermore, the biological origin of this reagent poses ethical issues, particularly with regard to animal protection and suffering, as well as problems of stock depletion leading to shortages [11]. Thus, the contribution of the FCS to culture media seems necessary, but also controversial. For these reasons, the common use of serum-free synthetic media would enhance the homogeneity of experiments by increasing the level of standardization, thus improving the quality of cell modeling worldwide [12].

This discipline is using more and more three-dimensional (3D) cell culture models, as they reproduce more precisely the tumor microenvironment. Indeed, 3D cell cultures produced by liquid overlay or scaffold techniques, allow the formation of pseudo-microtissues also called spheroids or organoids [13,14]. These structures are able to produce their own cell-to-cell and cell-to-matrix interactions, leading to biomimicking genetical, biochemical and phenotypical gradients. These methods are long to develop, but extremely advantageous, especially for oncology in vitro preclinical modeling being intensively promoted in order to reduce, refine or replace animal experimentation [15]. Thus, the mastering of their reproducibility in terms of growth characteristics is essential for efficient drug screening and efficacy prediction [14,16,17,18].

For this research, two liquid-overlay 3D cell culture types already developed by our team for drug screening (i) the proliferative MDA-MB-231 spheroid model and (ii) the non-proliferative SUM1315 spheroid model were used. This modeling highlighted that the choice of a proliferative or non-proliferative 3D model seems to be decisive for drug efficacy’s prediction, according to drug’s mechanism of action, i.e., cytotoxic or cytostatic [19]. In this context, to optimize the predictive responses of both 3D cell culture models to several chemotherapeutics, these works were focused on the study of SUM1315 and MDA-MB-231 cell lines response in a new synthetic serum-free medium, OptiPASS (BIOPASS). For this, several cell parameters, such as morphology, metabolic activity, viability/mortality, and sensitivity to anticancer drug, were analyzed in 2D or 3D cell culture conditions with OptiPASS medium and with their reference-serum-supplemented media.

## 2. Experimental Section

### 2.1. Maintenance of Cell Cultures

The triple-negative breast cancer cell lines (TNBC) MDA-MB-231 (ATCC HTB-26) and SUM1315 (MO2, ASTERAND) were maintained in a humidified incubator at 37 °C under 5% CO_2_. Cell culture was processed in 75 cm^2^ culture flasks (Falcon, Corning, NY, USA) with 15 mL of the appropriate culture medium. For culture of MDA-MB-231 line, the reference medium was the RPMI 1640 medium (Thermo Fisher Scientific, Waltham, MA, USA) supplemented with fetal calf serum (FCS, 10%, Eurobio, Paris, France) and gentamycin (20 μg/mL, Panpharma, Luitre, France). For the SUM1315 cell line, the reference medium was the Ham’s F-12 medium (Thermo Fisher Scientific), supplemented with 5% FCS, 10 mM HEPES buffer (SIGMA), insulin (4 μg/mL, Novo Nordisk, Bagsværd, Denmark A/S) and epidermal growth factor (EGF, 10 ng/mL, SIGMA). These two respective media were labeled “reference” in experiments with both cell lines. For experiments with OptiPASS medium (BIOPASS), the latter was supplemented with gentamycin (20 μg/mL) and was labeled “OptiPASS” in our experiments. Cells were only cultured before passage number 30 to prevent cell line drift.

### 2.2. 2D Cell Culture Development

#### 2.2.1. Cell Seeding

For both cell lines, cells were seeded at least in triplicates for each culture condition, at concentrations of 2 × 10^4^ cells per cm^2^ in 24-well microplates or 6-well microplates (Falcon) in 2 or 3 mL of culture medium, respectively. For cultures with OptiPASS medium, prior to seeding, wells were coated for 1 h under agitation with 2% Geltrex LDEV-free (Thermo Fisher Scientific, A14132-02) diluted in medium.

#### 2.2.2. Cell Counting with Cytation3MV Plate Reader and Trypan Blue

The cellular carpet of each microplate wells was imaged daily with the Cytation3MV plate reader (BioTek, Winooski, VT, USA; M = 4X) coupled to Gen 5.03 software until cells have reached the confluence (day seven). The number of living-adherent cells was then calculated for each culture condition by the “*cellular analysis*” algorithm of Gen 5.03 software, with the following settings: threshold value = 1200, background = light, min object size = 15 µm, max object size = 50 µm. The proliferation ratio was calculated by dividing the cell number of each well at day X by the number of cells one day after seeding (day one).

Then, when cells were at confluence, the supernatant containing floating cells (living and dead) was collected. Adherent cells were trypsinized and also collected, allowing the determination of “adherent”, “floating” and “dead” cells proportion of each cell culture condition, by a trypan blue exclusion test.

#### 2.2.3. Metabolic Activity Monitoring with Resazurin Test

Cell metabolic activity was determined by the resazurin test after one, three, five and seven days of culture (confluence) in reference or OptiPASS medium. It is a colorimetric assay involving the reduction of resazurin (non-fluorescent) into resorufin (fluorescent, *ex/em 570/584 nm*) by the metabolically active cells. For this, the cellular carpet of each condition was rinsed in hot D-PBS (37 °C) and incubated with 100 µL of 60 µM resazurin (ACROS Organics, Geel, Belgium). The quantity of resorufin formed was monitored with Cytation3MV plate reader and presented in the graphs as the metabolic activity of cells ± s.d (standard deviation).

#### 2.2.4. Expression of the Mesenchymal Marker Vimentin by Immunofluorescence

SUM1315 and MDA-MB-231 cells cultured with reference or OptiPASS medium were seeded in Ibitreat microplates (IBIDI) at concentrations of 5 × 10^5^ cells per well. For cells cultured with OptiPASS medium, a 1 h-coating with 2% Geltrex (in medium) was realized prior seeding. Twenty-four hours after seeding, cells were fixed for 20 min with a solution of 4% paraformaldehyde (Sigma-Aldrich, St. Louis, MO, USA) and permeabilized in ethanol 70° at −20 °C for 5 min. Cells were incubated for one hour with a saturation solution (0.1/1% v/w Triton/BSA in PBS) and then for one hour with the primary antibody (Vimentin, mouse, isotype IgG1a, 1/250 in PBS 0.1% Triton, Thermo Fisher Scientific) or the isotype control (1/1000, isotype IgG1a mouse, Thermo Fisher Scientific). Then, cells were incubated with the anti-mouse secondary antibody (1/400, AlexaFluor 647, Thermo Fisher Scientific) for one hour, and then with 10 µg/mL Hoechst 33258 (Sigma) for nuclear counterstaining. Finally, cells were dried, mounted with ProlongDiamond (Thermo Fisher Scientific) and observed with Cytation3MV plate reader (Dapi and Cy5 cube fluorescent filter cubes). The vimentin expression per cell was quantified from the immunofluorescent staining images obtained with both cell lines and culture conditions. For this, the fluorescent intensity of vimentin was normalized on the cellular positive area of each image (Cytation3MV).

### 2.3. 3D Cell Culture Development

#### 2.3.1. Breast Cancer Spheroids Production by the “Liquid Overlay” 3D Technique

3D cell culture protocol with TNBC cell lines was processed according to Dubois et al. [19]. Briefly, cells cultured in reference or OptiPASS medium, respectively, were seeded in 96-well “ultra-low-attachment” microplates (Corning, Corning, NY, USA; 4520) at a concentration of 1000 cells per well and left in the incubator to let the cells aggregate at the bottom of the wells. After 24 h of incubation, a cold solution (4 °C) of Geltrex LDEV-free (Thermo Fisher Scientific; A14132-02) diluted in the appropriate culture medium (reference or OptiPASS) was dispensed in each well to a final concentration of 2%. Microplates were then agitated for 20 min at 185 rpm and left in the incubator for 24 h. After this incubation time, unique compact spheroids were formed with reproducibility of spheroid formation per plate superior to 98% for both culture conditions. The level of spheroid compaction was determined by the establishment of a correlation between the number of seeded cells and the size of spheroids formed (Figure A1, Appendix A, example for MDA-MB-231). In this experiment, the spheroids presented similar and high opacity in bright field imaging for all cell seeding concentrations. This methodology was used to estimate the compaction level of spheroids in all experiments. 

#### 2.3.2. Characterization of Spheroid Size and Morphology with Cytation3MV

Spheroids formed with the two culture conditions were imaged daily (M = 4X) from day one (day of effective spheroid formation) to day 14, with Cytation3MV coupled to Gen 5.03 software. The size of each spheroid was measured with the “*cellular analysis*” algorithm of the software with the following settings: threshold value = 10000, background = light, min object size = 180 µm, max object size = 1000 µm.

#### 2.3.3. Spheroid Viability Monitoring with Live/Dead Fluorescent Probes

Spheroid viability was monitored after seven and fourteen days of culture (day 7, day 14) by the live/dead kit (Molecular Probes, L3224), labeling viable cells in green (calcein-AM penetration) and dead cells in red (ethidium homodimer-1 penetration, etdh-1). For this, spheroids were harvested and deposited into six-well microplates. Spheroids were rinsed twice with D-PBS (37 °C) and incubated with a solution of 2 µM calcein-AM and 4 µM etdh-1 in D-PBS for 45 min (protected from light), according to supplier’s instructions. Spheroids were then transferred into eight-well uncoated micro slides (IBIDI, 80821) and imaged on Cytation3MV plate reader with green fluorescent protein cube (GFP) and Propidium Iodide (PI) fluorescence cube loaded on the device (M = 4X).

#### 2.3.4. Spheroid Anticancer Drug Sensitivity Determination

Spheroids formed in reference medium or OptiPASS medium were treated with 1 µM of epirubicin (Carbosynth, Compton, UK; stock solution of 1 mM dissolved in distilled water), for five days. The size of control and 1 µM-treated spheroids was monitored with Cytation3MV for five days, as previously described. The percentage of treated spheroids cell viability at day five was determined with resazurin test, by the ratio of resorufin formed in treated spheroids on control spheroids. These values were then normalized on the percentage of spheroids treated size on control size, at day five. Finally, the viability assay live/dead was performed on treated and control spheroids at day five. The quantification of ethidium homodimer-1 was realized on epirubicin-treated spheroids by the ratio of Etdh1 intensity in treated spheroids normalized Etdh1 intensity in control spheroids (Cytation3MV).

### 2.4. Statistical Analysis

Results were expressed as means ± standard deviation (s.d.) or standard error (s.e.m) of *n*-independent experiments. Every experiment was performed at least in triplicate and then statistically compared using a two-sided student’s t-test. Data comparison was considered statistically significant when the p-value was p < 0.05 (*). Lower p-values were noted in figures as follows: p < 0.01 (**), p < 0.001 (***), p < 0.0001 (****) and p < 0.00001 (*****). Non-significant results were noted as “*ns*”.

## 3. Results

### 3.1. Monolayer (2D) Culture Development of TNBC Cell Lines with OptiPASS Medium

In this study, the performances of the serum-free medium OptiPASS were first determined in 2D cell culture on two TNBC cell lines, MDA-MB-231, and SUM1315. For this, cells were cultured with their reference medium (containing 5 to 10% of FCS, according to cell line) or OptiPASS medium (coated with 2% Geltrex), until confluence. In order to characterize cell behavior, several parameters, such as (i) cell adherence, (ii) cell proliferation, (iii) cell metabolic activity, and (iv) cell morphology, were analyzed in both cell culture media. Additionally, the mesenchymal phenotype status of both cell lines was also determined in each culture medium by vimentin immunostaining (Figure 1 and Figure 2).

For MDA-MB-231 cell line, the cell proliferation rate was first studied in both culture media by a cell counting experiment using the Cytation3MV plate reader. For this, the ratio of adherent cells at day X on adherent cells at day one (one day after seeding) was calculated (Figure 1a). The proliferation level increased from 1.7 ± 0.1 and 1.7 ± 0.1 at day three (*p* = 0.71, two-sided student’s t-test), to 3.4 ± 0.8 and 3.0 ± 0.2 at day seven (p = 0.05) for reference medium and OptiPASS medium, respectively. The cell proliferation time course seemed to be similar in reference and OptiPASS medium. In parallel, cell metabolic activity was monitored for the seven days of culture by the resazurin test. It was 0.066 ± 0.007 and 0.052 ± 0.005 at day one (p = 0.0003), 0.181 ± 0.039 and 0.084 ± 0.006 at day three (p = 0.0005) and increased to 0.335 ± 0.078 and 0.366 ± 0.056 at day seven (p = 0.40), for reference and OptiPASS medium, respectively (Figure 1b). Then, the cell repartition profiles (either adherent to the support, floating in the supernatant or dead in the supernatant) were analyzed for both cell culture conditions with a blue trypan exclusion test at confluence (at day seven) for reference and OptiPASS medium (Figure 1c). The proportions of MDA-MB-231 cells attached to the support in reference and OptiPASS medium were similar with 78 ± 8% and 74 ± 11% (p = 0.64), respectively. Additionally, the proportions of floating-living cells in reference medium were not significantly different to OptiPASS medium. Indeed, it was of 15 ± 9% in reference medium and of 8 ± 5% (p = 0.30) in OptiPASS medium. Finally, the rate of dead cells in reference medium was of 7 ± 3% at confluence and similar to OptiPASS with 18 ± 11% (p = 0.20). These results showed that the proportions of “adherent”, “floating” and “dead” cells in OptiPASS medium were similar to reference medium. Thus, the morphology of the cellular carpet observed in digital phase contrast for the seven days of culture showed no difference between cells cultured in OptiPASS medium compared to reference medium (Figure 1d). Finally, vimentin immunostainings analysis carried out on MDA-MB-231 cell line (Figure 1e,f) showed the constant expression of the mesenchymal marker in cells cultured with both culture media. Indeed, the vimentin expression in MDA-MB-231 cells was of 8.2 ± 0.2 × 10^5^ AU and 8.8 ± 0.7 × 10^5^ AU in reference and OptiPASS medium, respectively (p = 0.27). These results demonstrated that for all analyzed parameters, i.e., cell proliferation rates, cell metabolic activity and the proportion of attached cells/floating/dead cells, similar cell culture performances were detected for OptiPASS and reference media, with MDA-MB-231 cell line.

For SUM1315 cell line, the cell proliferation analysis showed growth rates of 1.9 ± 0.2 at day three, 3.3 ± 0.4 at day seven in reference medium and of 1.2 ± 0.1 at day three (p = 10^−6^ compared to reference) and 1.9 ± 0.3 at day seven (p = 10^−12^ compared to reference), in OptiPASS medium (Figure 2a), respectively. Similarly, the cell metabolic activity was determined in the same experimental conditions and was of 0.217 ± 0.016 in reference medium and 0.057 ± 0.004 in OptiPASS medium (p = 10^−9^) at day three. Then, it increased at day seven for both cell culture media with 0.148 ± 0.019 and 0.067 ± 0.014 (p = 10^−6^), for reference and OptiPASS, respectively (Figure 2b). Then, the proportion of adherent cells, living-floating cells, and dead-floating cells was analyzed in each cell culture medium at confluence (at day seven) (Figure 2c). The proportion of adherent cells in reference medium was of 66 ± 12%. In contrast, with OptiPASS medium, it was lower with 22 ± 14% (p = 0.01 compared to reference) (Figure 2c). Conversely, the rate of floating-living cells remained lower in reference medium with 25 ± 10% compared to 74 ± 15% in OptiPASS (p = 0.01) (Figure 2c). Interestingly, no significant difference in the rate of dead cells was detected between reference and OptiPASS medium with 9 ± 7% and 4 ± 1% (p = 0.37), respectively (Figure 2c). Then, the observations of cell morphology in digital phase contrast (Figure 2d) showed a majority of homogeneously spread and attached cells in reference medium. In contrast, a cluster of round-detached cells in the middle of the wells was detected with OptiPASS medium (Figure 2d, magnification). Nevertheless, the immunostaining experiment confirmed the maintained expression of the vimentin marker with both cell culture conditions (Figure 2e), demonstrating that SUM1315 cells cultured in OptiPASS medium had kept their mesenchymal phenotype. Indeed, the vimentin expression was of 8.3 ± 0.04 10^5^ and 8.4 ± 0.5 10^5^ AU per cell, respectively (p = 0.79) (Figure 2f). For SUM1315 cell line, all studied parameters showed a greater performance in terms of cell proliferation and cell metabolic activity with reference medium compared to OptiPASS medium. However, no difference in cell mortality was detected in both culture conditions (Figure 2c). Thus, the cellular behavior of SUM1315 cell line was different in OptiPASS medium. Indeed, in this latter, a higher proportion of living-floating cells was detected suggesting that despite the weak attachment of cells in this medium, they remained alive over time. Similarly, the cell proliferation rate of adherent cells remained positive, i.e., superior to one and the mesenchymal phenotype of cells was clearly maintained in OptiPASS medium.

### 3.2. Three-Dimensional (3D) Cell Culture of TNBC Cell Lines with OptiPASS Medium

#### 3.2.1. Spheroid Development in OptiPASS Medium

OptiPASS medium performances were then compared to reference medium for the development of 3D cell cultures with MDA-MB-231 and SUM1315 triple-negative breast cancer cell lines. For these experiments, cell lines previously cultured in monolayer with reference or OptiPASS medium were harvested and cultured in 3D condition according to Dubois et al. [19] with their respective destination medium. In these cell culture conditions, spheroid size monitoring was analyzed for 14 days of culture with Cytation3MV plate reader (BIOTEK). In parallel, spheroid viability was assessed with Live/Dead fluorescent labeling at day seven and day 14.

For MDA-MB-231 cell line, the mean spheroid sizes were of 339 ± 85 µm and 422 ± 102 µm at day two (p = 0.0002) and increased to 856 ± 117 µm and 1137 ± 120 µm (p = 10^−17^) at day eight for reference and OptiPASS medium, respectively (Figure 3a). Then, spheroid diameter reached a growth plateau at day 14 with 1011 ± 73 µm and 1183 ± 80 µm at day 14 (p = 10^−13^), for reference and OptiPASS medium, respectively (Figure 3a). With this cell line, the mean spheroid sizes with OptiPASS medium always remained superior to reference medium. Then, spheroid viability was assessed with live/dead kit (Molecular probes), staining viable cells in green (Calcein-AM penetration) and dead cells in red (ethidium homodimer-1 penetration). For both cell culture conditions at day seven, spheroids exhibited a majority of green staining (Figure 3b). Then, at day 14, a core of dead cells (red labeling), surrounded by a proliferative cell ring (green labeling of viable cells) was detected. In parallel, spheroid topology and level of compaction remained constant throughout the experiment, whatever the cell culture condition (Figure 3b). These results demonstrated the same spheroid viability/mortality profiles between reference medium and OptiPASS medium, for MDA-MB-231 cell line.

Similarly, for SUM1315 cell line, the mean spheroid sizes at day two were of 292 ± 21 µm and 319 ± 34 µm (p = 0.02) and increased to 476 ± 51 µm and 711 ± 77 µm at day eight (p = 10^−35^) and to 585 ± 82 µm and 961 ± 81 µm at day 14 (p = 10^−44^) in reference and OptiPASS medium, respectively (Figure 3c). Then, for both cell culture conditions, live/dead fluorescent analysis showed similar spheroid green staining (viable cells) after seven days of culture (Figure 3d). This profile then changed after 14 days, with the presence of a visible dead core in spheroids (red staining) surrounded with viable cells, for reference and OptiPASS medium conditions (Figure 3d). Additionally, the same spheroid compaction and topology were found in SUM1315 spheroids cultured in OptiPASS medium compared to reference medium. As previously, spheroids formed with OptiPASS medium clearly presented sizes superior to reference medium conditions. More, similar viability/mortality profiles were also detected between spheroids cultured in OptiPASS compared to reference.

#### 3.2.2. Spheroid Sensitivity to Drugs in OptiPASS Medium

Then, spheroid sensitivity to the anticancer agent epirubicin was studied in OptiPASS medium in comparison to reference medium (Figure 4). For this, spheroids of MDA-MB-231 and SUM1315 cell lines were treated with 1 µM epirubicin for five days. First, the evolution of spheroid size was monitored for the five days of treatment (Figure 4a,b). Then, resazurin viability test (Figure 4c,d) and Live/Dead fluorescent analysis (Figure 4e,f) and ethidium-homodimer-1 intensity quantification in treated-spheroids (Figure 4g,h) were performed at the end of the treatment on the 1 µM epirubicin-treated spheroids in comparison with control spheroids.

In reference medium, MDA-MB-231 spheroid size increased from 330 ± 16 µm (at day 0) to 659 ± 11 µm (at day five, p = 10^−12^) (Figure 4a). After treatment with 1 µM epirubicin, it increased slightly to 402 ± 14 µm after three days of treatment (p = 0.004, compared to day zero), and then decreased significantly to 297 ± 62 µm (p = 0.0002, compared to day three). With OptiPASS medium, control spheroid size increased significantly from 491 ± 8 µm at day zero to 855 ± 53 µm at day five (p = 10^−34^). In contrast, epirubicin-treated spheroids size increased slightly to day three with 542 ± 25 µm (p = 0.04, compared to day zero) and remained stable until day five (p = 0.84). Thereby, for the five days of 1 µM epirubicin treatment, spheroids from both cell culture medium conditions showed significantly decreased sizes compared to their respective untreated controls. This highlights the cytostatic activity of epirubicin in these both cell culture conditions. Then, the percentage of spheroid viability (normalized to spheroid size) was determined after five days of treatment with epirubicin (Figure 4c), by the ratio of treated spheroids metabolic activity on untreated spheroids activity. For MDA-MB-231 cell line, the percentages of treated spheroids viability were similar between the two culture medium conditions, with 67 ± 5% for reference spheroids and 62 ± 6% for OptiPASS spheroids (p = 0.15) (Figure 4c). Finally, live/dead labeling was processed on both control and treated spheroids after five days of treatment (Figure 4e). The results showed that epirubicin-treated spheroids cultured in reference medium presented overall green staining (with 2 ± 1% of ethidium-homodimer positive cells compared to controls, Figure 4f), demonstrating the presence of mainly viable cells composing the spheroids. In contrast, the analysis of treated spheroids cultured in OptiPASS medium showed clearly an increase in dead cells (57 ± 22% of Etdh1+ cells, Figure 4f), revealing also a higher cytotoxic response with this chemotherapeutic agent.

In parallel, SUM1315 spheroids sensitivity to epirubicin was determined in OptiPASS medium and compared to reference medium. Control SUM1315 spheroids size did not increase in reference medium for the five days of treatment with 197 ± 14 µm at day zero and 227 ± 8 µm at day five (p = 0.07) (Figure 4b). Similarly, epirubicin-treated spheroids size remained stable for the five days of culture with 219 ± 9 µm at day five (p = 0.21 compared to day 0). Additionally, no significant difference in spheroids size was detected between epirubicin treated spheroids and control, in reference medium (p = 0.13). In contrast, with OptiPASS medium, SUM1315 spheroids size increased from 235 ± 6 µm at day zero to 319 ± 6 µm at day five (p = 10^−14^). In parallel, epirubicin treated spheroids size increased until day three with 263 ± 7 µm (p = 0.02, compared to day 0), but remained stable until day five (272 ± 11 µm, p = 0.91 compared to day three). Thus, with this cell line, a significant difference in spheroid size between untreated and treated conditions was detected at day five, only when spheroids were cultured in OptiPASS medium (p = 0.0001). This showed the cytostatic action of epirubicin on SUM1315 spheroids only with OptiPASS medium. Then, the analysis of spheroid viability after five days of treatment showed similar percentages of the viability of 84 ± 8% and 91 ± 10% (p = 0.23), for reference treated spheroids and OptiPASS treated spheroids, respectively (Figure 4d). Finally, Live/dead images (Figure 4f) and Etdh1+ quantification (4 ± 0.5% and 7 ± 5%, p = 0.28, Figure 4h) showed similar viability and mortality patterns between reference and OptiPASS culture conditions, and this for control and epirubicin-treated spheroids.

## 4. Discussion

Preclinical studies aim to screen new drugs or new treatment strategies. These consist of several steps, including in vitro cell culture modeling. For this, the culture medium plays a crucial role for the metabolism and the growth of cells in culture. As a result, the stability and homogeneity of culture medium components are essential factors for ensuring the proper cell response to drugs in in vitro screenings [20,21,22]. Otherwise, culture media might need the adding of animal-serum in their composition, in order to ensure and maintain cell development and homeostasis. Nevertheless, the use of serum is controversial in the cell culture field, principally because of its heterogeneity. The challenge then appears to be the removal of serum from media, by using new synthetic formulations. In this context, the objective of this study was to determine the performance of a new synthetic serum-free medium OptiPASS (BIOPASS) adapted for the culture of mesenchymal stem cells, whose formulation is an industrial secret. These works were processed in monolayer and 3D cell culture conditions, using these TNBC cell culture models cultured in OptiPASS medium or their serum-containing-reference media, RPMI1640 medium (supplemented with 10% FCS) and Ham’s F12 medium (supplemented with 5% FCS), respectively.

Firstly, OptiPASS medium performances were determined in monolayer cell culture by the analysis of cell adherence, cell proliferation, and cell metabolic activity. Then the expression of the mesenchymal phenotype marker vimentin was determined. Indeed, vimentin is the most commonly used marker for the determination of a phenotype change from epithelial to mesenchymal for various models, including breast cancers [16,23,24,25]. Thus, for OptiPASS cell culture development in our experimental conditions, it was necessary to add a coating step before cell seeding. The latter was carried out with the deposition of 2% Geltrex solution containing extracellular matrix proteins, in order to trigger cell adherence. Indeed, contrary to the serum which origin cannot be controlled, the Geltrex^®^/Matrigel^®^ are produced by EHS controlled cells, extracted and purified by an industrial process [26]. Thus, this Geltrex concentration was already determined and used for the development of 3D cell cultures for both MDA-MB-231 and SUM1315 cell lines [19]. With MDA-MB-231 cell line model, the same cell adherence and cell proliferation rates were detected during the experiment in RPMI1640 medium and in OptiPASS medium (at day 7, 78% vs. 74%, p = 0.64 and 3.0 vs. 3.4 p = 0.05, respectively). Nevertheless, MDA-MB-231 metabolic activity was firstly lower from day one to day three in OptiPASS medium compared to reference medium. This can be explained by a delayed attachment of the cells in OptiPASS medium that delays the restart of cells metabolic activity. Then, this parameter remained similar from day five until confluence (0.335 vs. 0.366 at day seven, p = 0.40). Similarly, the cellular expression of the mesenchymal marker vimentin remained comparable in cells grown in OptiPASS medium and reference medium, showing the absence of cell line drift with this culture condition. These results suggest that OptiPASS medium is suitable for the monolayer culture of MDA-MB-231 cell line, providing the same performances as the reference medium. In contrast, with SUM1315 cell line, lower (i) cell adherence (66% vs. 22%, p = 0.01), (ii) cell proliferation rates (3.3 vs. 1.9, p = 10^−12^) and (iii) cell metabolic activity (0.148 vs. 0.067, at day seven, p = 10^−6^) were detected in reference medium compared to OptiPASS medium, respectively. Additionally, SUM1315 cells mostly presented an ovoid shape in bright field microscopy. These results suggest that cells cultured with OptiPASS medium may partly not adhere to the Geltrex coating or may detach. This may cause the decreased values of cell proliferation rates and metabolic activity for this cell line, as the cells may have detached prematurely. SUM1315 mesenchymal phenotype was still maintained in OptiPASS medium as depicted with the vimentin staining [27]. Besides, in OptiPASS medium, the SUM1315 cell proportion profile revealed the presence of a majority of viable cells in the supernatant (74% for OptiPASS vs. 25% for reference at day seven, p = 0.01). All these results showed that OptiPASS medium was adapted for the expansion of MDA-MB-231 and SUM1315 cell lines cultured in monolayer, allowing to proceed to further 3D cell culture production steps.

Since a few decades, new cell cultures models in three-dimensions (3D cell cultures) have been dramatically developing, as they reproduce more closely tumor’s microenvironment [28,29], and are increasingly included in the drug development pipelines of preclinical studies [13,30]. Thus, our team already developed 3D cell cultures on TNBC cell lines, highlighting two different models i.e., SUM1315 non-proliferative and MDA-MB-231 proliferative spheroids, in their respective serum-containing reference media [19]. In this context, OptiPASS performances were tested on these 3D specific models. For this, spheroid size monitoring, spheroid viability/mortality monitoring and spheroid sensitivity to anticancer drugs were assessed on both cell lines cultured in OptiPASS and compared to reference medium. First, the analysis of spheroid size evolution showed a 1.3 and 1.5-fold increase in spheroid size when cultured with OptiPASS compared to reference medium, for MDA-MB-231 and SUM1315 cell lines, respectively. Then, the viability/mortality fluorescent experiments showed the same profiles between OptiPASS and reference-produced spheroids. Additionally, the formation of a necrotic core surrounded by a ring of viable/proliferating cells was clearly observed at day 14 for both culture conditions. This might be explained by the formation of nutrient and oxygen gradients within the spheroids, thus reproducing the in vivo behavior of tumors, as already described in the literature [17]. All of these results demonstrated that OptiPASS medium is very suitable for the 3D cell culture of TNBC cell lines MDA-MB-231 and SUM1315. Furthermore, it promoted enhanced spheroid proliferation rates, especially for SUM1315 cell line, turning this cell line into a high proliferative model in this culture condition. Indeed, SUM1315 is a highly heterogeneous and undifferentiated cell line (claudin-low phenotype) [31] giving it a low proliferation rate in 2D (doubling time > 72 h). Thus, the evolution of spheroid size in reference medium was minimal at long-term. In contrast, the culture of SUM1315 spheroids in OptiPASS allowed a high increase in spheroid proliferation rate. This last property allows the detection of both cytostatic and cytotoxic drug activity, where appropriate.

In order to use 3D cell cultures produced in OptiPASS medium for therapeutic drug screenings, spheroid sensitivity to epirubicin was determined in OptiPASS and compared to the reference medium. Spheroids were treated with epirubicin, a strong intercalating agent used in conventional chemotherapeutics protocols, presenting high cytostatic (slowed or stopped proliferation rate) and cytotoxic (decrease in viability and/or presence of dead cells) activities. For MDA-MB-231 spheroids treated with 1 µM epirubicin in reference medium, a 2.2-fold decrease in size, as well as a decrease in viability (67%), were detected compared to controls. Similarly, MDA-MB-231 spheroids cultured in OptiPASS medium presented a 1.6-fold size decrease after treatment associated with lowered viability (62%, similar to reference). Therefore, for this cell line, the cytostatic and cytotoxic actions of epirubicin were detected whatever the culture condition (reference or OptiPASS). In contrast, for SUM1315 spheroids cultured in reference medium, no difference in spheroid size was noted after the treatment, but only a viability drop (84% viability). Conversely, SUM1315 spheroids cultured in OptiPASS presented a 1.2-fold size spheroid decrease with slowed viability (91%, similar to reference). These results showed that the culture with OptiPASS medium allowed to detect the epirubicin cytostatic and cytotoxic actions on both cell lines.

All these results demonstrated the added-value of OptiPASS synthetic medium, for the TNBC 3D cell cultures development. Indeed, cell proliferation, cell viability and cell drug sensibility in MDA-MB-231 and SUM1315 spheroids models were clearly optimized. These results also highlighted the key role of the culture medium in the optimization of culture models’ predictive response within the drug screening pipeline and opened the way to the common use of this medium in preclinical breast cancer research studies. Other experiments aiming to determine OptiPASS performances with other cancer cell lines are currently underway.

## Figures and Tables

**Figure 1 jcm-08-00397-f001:**
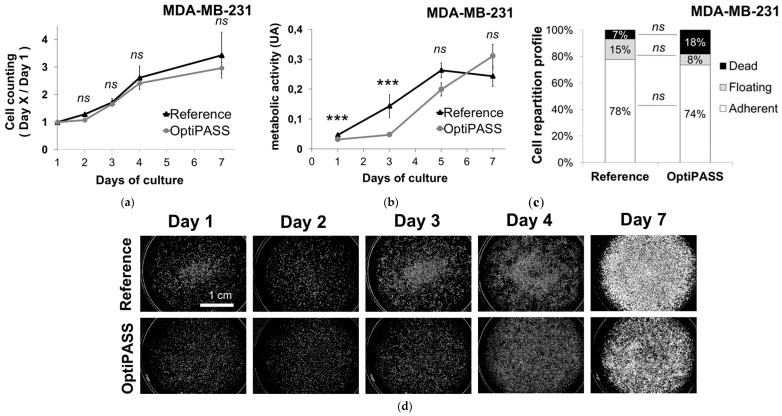
MDA-MB-231 monolayer cell culture development with OptiPASS medium. (**a**) 2D proliferation rates: the number of adherent cells was counted with Cytation3MV plate reader (BIOTEK) after one, two, three, four and seven days of culture and normalized to cell counting at day one, n = 9, bars = s.e.m. (**b**) Cell metabolic activity monitoring: the metabolic activity of adherent cells was quantified with resazurin test for seven days and normalized to cell metabolic activity quantified at day one, n = 16, bars = s.d. (**c**) Cell repartition profile: cells cultured in reference medium or OptiPASS medium were harvested and counted with a blue trypan exclusion test. Three cell categories are represented: “floating” and “dead” cells were counted in the supernatant medium, and “adherent” cells were counted after trypsinization of the cellular carpet. n = 5, ns = non-significant difference with student *t*-test (**d**) Cellular carpet morphology in phase contrast imaging: each well of the six-well microplate was imaged at day one, two, three, four and seven (confluence) with Cytation3MV plate reader. Images (M = 4X) were stitched to allow an overview of the all well. Well area = 9.6 cm^2^. (**e**)Vimentin immunofluorescent staining: cells cultured in reference or OptiPASS medium were immunoassayed for vimentin expression and imaged with Cytation3MV (GFP and Cy5 filters). The isotype control is the negative control for vimentin expression. Scale bar = 100 µm, blue = nuclei, yellow = vimentin. (**f**) Vimentin intensity quantification per cell (AU, error bars = s.d.). ns = non-significantly different, *** = p < 0.001, two-sided student’s *t*-test comparison.

**Figure 2 jcm-08-00397-f002:**
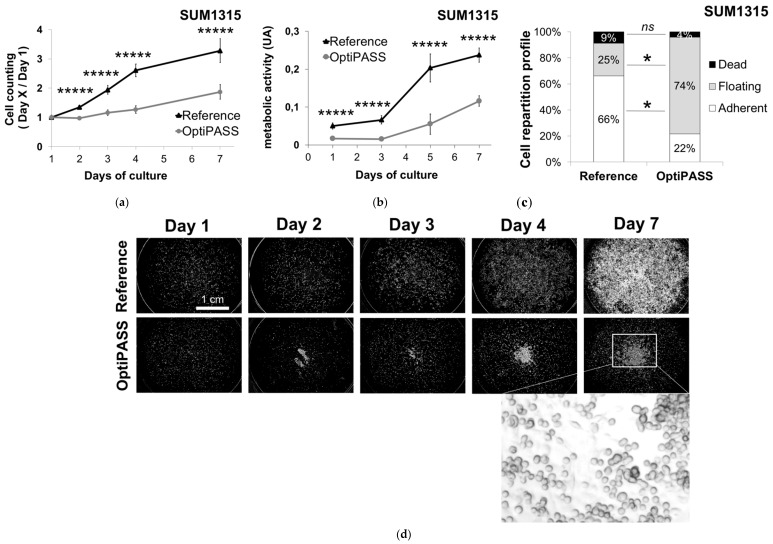
SUM1315 monolayer cell culture development with OptiPASS medium. (**a**) 2D proliferation rates: the number of adherent cells was counted with Cytation3MV plate reader (BIOTEK) after one, two, three, four and seven days of culture and normalized to cell counting at day one, n = 9, bars = s.e.m. (**b**) Cell metabolic activity monitoring: the metabolic activity of adherent cells was quantified with resazurin test for seven days and normalized to cell metabolic activity quantified at day one, n = 16, bars = s.d. (**c**) Cell repartition profile: cells cultured in reference medium or OptiPASS medium were harvested and counted with a blue trypan exclusion test. Three cell categories are represented: “floating” and “dead” cells were counted in the supernatant medium, and “adherent” cells were counted after trypsinization of the cellular carpet. n = 5, ns = non-significant difference with student *t*-test (**d**) Cellular carpet morphology in phase contrast imaging: each well of the six-well microplate was imaged at day one, two, three, four and seven (confluence) with Cytation3MV plate reader. Images (M = 4X) were stitched to allow an overview of the all well. Well area = 9.6 cm^2^. (**e**) Vimentin immunofluorescent staining: cells cultured in reference or OptiPASS medium were immunoassayed for vimentin expression with Cytation3MV (GFP and Cy5 filters). The isotype control is the negative control for vimentin expression. Scale bar = 100 µm, blue = nuclei, yellow = vimentin. (**f**) Vimentin intensity quantification per cell (AU, error bars = s.d.). ns = non-significantly different, ***** = p < 0.00001, two-sided Student’s *t*-test comparison.

**Figure 3 jcm-08-00397-f003:**
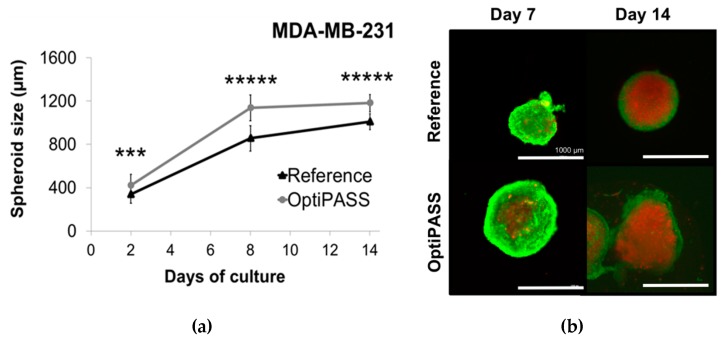
MDA-MB-231 (**a**,**b**) and SUM1315 (**c**,**d**) spheroid development with OptiPASS medium. (**a**,**c**) Spheroid size monitoring: spheroids formed with reference and OptiPASS medium were imaged after 2, 8 and 14 days of culture with Cytation3MV (BIOTEK). The mean size of each spheroid condition (n = 48) was calculated by the “cellular analysis” algorithm of Gen5.03 software (BIOTEK), bars = s.e.m. (**b**,**d**) Spheroids viability/mortality profile: after 7 and 14 days of culture, spheroids (n = 5) were harvested and incubated with live/dead fluorescent probes. Green signals correspond to Calcein-AM penetration in viable cells, and red signals correspond to ethidium homodimer-1 penetration in dead/necrotic cells. Observations were made with Cytation3MV with focal height fixed to the center of each spheroid (GFP and Texas Red filters, objective = 10×). * = p < 0.05, *** = p < 0.001, ***** = p < 0.00001, two-sided student’s *t*-test comparison.

**Figure 4 jcm-08-00397-f004:**
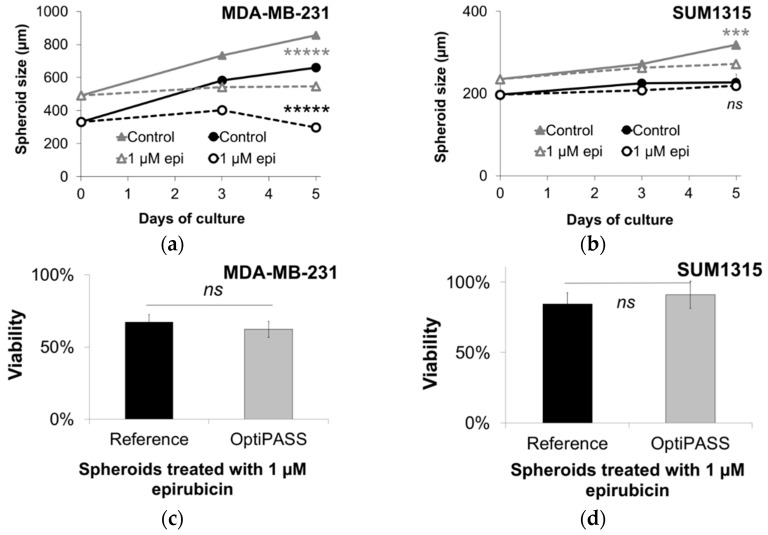
MDA-MB-231 (**a**,**c**,**e**,**g**) and SUM1315 (**b**,**d**,**f**,**h**) spheroids sensitivity to epirubicin in OptiPASS medium. Spheroids formed with reference or OptiPASS medium were treated with 1 µM of epirubicin for five days. (**a**,**b**) Spheroid size monitoring after epirubicin treatment: the mean size of control and treated spheroids was measured for the five days of treatment with Cytation3MV (BIOTEK), n = 6, bars = s.d. (**c**,**d**) Spheroid viability after epirubicin treatment: the percentage of treated spheroids viability was determined by the ratio of metabolic activity of treated spheroids on untreated spheroids at day five. Then, the viability was normalized on the percentage of treated spheroid size (on control spheroid size) calculated at day five from each cell culture medium, n = 6, bars = s.e.m. (**e**,**f**) Spheroid viability/mortality profile after epirubicin treatment: after five days of treatment, control and treated spheroids were incubated with Calcein-AM and ethidium-homodimer-1 (Etdh1+) (Live/dead kit). green = viable cells, red = dead cells. Observations were made with Cytation3MV with focal height fixed to the center of each spheroid. Scale bar = 1000 µm. (**g**,**h**) mean percentage of Etdh1+ staining in treated spheroids compared to control spheroids (error bars = s.d.). ns = non-significantly different, *** = p < 0.001, *****= p < 0.00001, two-sided Student’s *t*-test comparison.

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
