# Peer review of "The New Synthetic Serum-Free Medium OptiPASS Promotes High Proliferation and Drug Efficacy Prediction on Spheroids from MDA-MB-231 and SUM1315 Triple-Negative Breast Cancer Cell Lines"

_jcm, 2019, doi:10.3390/jcm8030397_

Reviewer 1 Report

The presented manuscript describes the use of a commercially available medium and supplement combination, OptiPASS, to culture triple-negative breast cancer cell lines, MDA-MB-231 and SUM1315. Both cell lines were grown in 2D and 3D-spheroid culture conditions in both the reference medium specific for each line and OptiPASS. Cell growth, metabolic activity, level of vimentin staining and cell morphology were observed to be similar in cells grown in OptiPASS compared with those cultured in reference medium in 2D culture conditions. Spheroid size and level of live/dead staining were observed to be similar when MDA-MB-231 cells were grown in 3D culture conditions in both reference and OptiPASS medium. SUM1315 cell spheroids, on the other hand, were observed to increase in size more rapidly and to become larger in OptiPASS compared to reference medium. Spheroids from each cell line in reference and OptiPASS medium were cultured with epirubicin to examine the predictive response of the culture method to finding cytostatic or cytotoxic drugs.

The manuscript is well written and easy to follow. The methods chosen were simple, but appropriate to show the required effects. It may have been nice to quantify how many cells expressed vimentin via flow cytometry instead of imaging alone. This quantification could even have been carried out with image analysis tools.

The methods section would be improved by referring to the temperature reagents were used at, instead of "hot" or "cold" as is the case now.

Line 148 states "After this incubation time, unique compact spheroids were formed with reproducibility superior to 98% for both culture conditions." With reference to this sentence what characteristic was reproducible?

Line 176 (and other places throughout the manuscript) change to "Student's t-test" not "student's t-test"

"Parallely" is used throughout the manuscript, but unfortunately is not a word. "In parallel" would be a better phrase to use. In addition, "more" is used throughout the manuscript, often to begin a sentence, this would be better written as "additionally".

The figures  appear to have fallen foul of some formatting issues in converting to pdf. There are however some issues that cannot be accounted for in this way. Please remove the boxes and lines from around the graphs. Smoothed lines are used in the line graphs. These are inappropriate in a scientific manuscript, please replace with lines directly between data points. The legends on the graphs, axis labels etc are very difficult to read, please make readable. 

Figures 1d, 3a, 3c, 4e, 4f do not have scale bars that are readable, please correct.

Figures 4 c, d please remove % from axis numbers, they are not needed.

In the legend for Figure 1 the following is stated: ***** = p<0.00001, however, *** is shown on the graph.

Figure 1b shows stastically significant differences  in metabolic activity between cells cultured in reference medium and OptiPASS, yet this is not highlighted or discussed.

Figure 2d shows low magnification images of SUM1315 cells grown in each medium. At this magnification and level of image reproduction it is difficult to see that there are "round-detached cells in the middle of the wells was detected with OptiPASS medium" as stated in line 230

In Figures 3a,c it is not stated whether images are maximum projection from confocal z-stacks or fluorescent images through the centre of spheroids. It would aid in understanding if this was clarified.

Lines 235-236 "More, similar cell repartition profiles were detected by the time (from seeding to confluence), for each culture medium." It is difficult to understand what this statement means. Please clarify the meaning.

In line 269, "level of compaction" is referred to. What is this measurement and how was it estimated/measured?

Lines 310-311 are written in such a way that suggests the drug is having a cytostatic effect on the media, not the cells growing within the media. Again, lines 365-368 should be rewritten from "media" to "cells grown in media"

In lines 315-321, fluorescent signals from live/dead staining are referred to as "markings", "images", "signals", "labelling" would all be more appropriate and scientific.

There was no real explanation as to why Geltrex was used or why it would be preferable to not have animal serum in the medium, but then to have to use an animal derived gel instead.

I would like to see some additional argument that it is acceptable to grow SUM1315 in OptiPASS medium. Especially since growing these cells in OptiPASS enhanced proliferation in spheroids, thus changing the model from non-proliferative to proliferative.

Author Response

To the editor:

Dear Ms. Rasic,  

 Thank you very much for your comments and complementary instructions (e-mail received on February the 28th, at the end of the document). Please find attached the revised manuscript untitled:

 The new synthetic serum-free medium OptiPASS promotes high proliferation and drug efficacy prediction on spheroids from MDA-MB-231 and SUM1315 Triple-Negative Breast Cancer cell lines.

By Clémence Dubois, Pierre Daumar, Corinne Aubel, Jean Gauthier, Bernard Vidalinc, Emmanuelle Mounetou, Frédérique Penault-Llorca& Mahchid Bamdad.

 According to your instructions, I applied point-by-point the modifications requested by the reviewers. Please see “Response to reviewer” below.

 Additionally, all the modifications have been done in the manuscript for each Reviewer in the "Track Changes" of Microsoft Word. All the figures have been modified according to reviewers’ instructions and one supplementary data (Figure S1) was added to the manuscript in the experimental section.

 I hope that our manuscript will be published in Journal of Clinical Medicine, I would be at your disposal for any further comment,

 Best regards,

 Mahchid Bamdad

Responses to reviewer:

1.    It may have been nice to quantify how many cells expressed vimentin via flow cytometry instead of imaging alone. This quantification could even have been carried out with image analysis tools.

 Response: We quantified the vimentin expression per cell based on the immunofluorescent staining images obtained from both cell lines and culture conditions. For this, we normalized the fluorescent intensity of vimentin on the cellular positive area of each image (results are expressed in Arbitrary Units) (Cytation3MV, Biotek). The results demonstrated a similar expression of this mesenchymal marker between the two culture conditions and both cell lines. Indeed, the vimentin expression in MDA-MB-231 cells was of 8.2±0.2 105 AU and 8.8±0.7 105 AU in reference and OptiPASS medium, respectively (p=0.27). Similarly, for SUM1315 cell line, it was of 8.3±0.04 105 and 8.4±0.5 105 AU per cell, respectively (p=0.79). These results were described in the manuscript and added in Figure 1g for MDA-MB-231 cell line and Figure 2g for SUM1315 cell line.

The following sentences were added in the manuscript:

 (Experimental section, lines 138-141): The vimentin expression per cell was quantified from the immunofluorescent staining images obtained with both cell lines and culture conditions. For this, we normalized the fluorescent intensity of vimentin on the cellular positive area of each image (Cytation3MV).

 (Caption, Figure 1 and Figure 2): (f) Vimentin intensity quantification per cell (AU, error bars = s.d.).

 (Results section, line 231-233): Indeed, the vimentin expression in MDA-MB-231 cells was of 8.2±0.2 105 AU and 8.8±0.7 105 AU in reference and OptiPASS medium, respectively (p=0.27).

(Results section, line 256-257): Indeed, the vimentin expression was of 8.3±0.04 105 and 8.4±0.5 105 AU per cell, respectively (p=0.79).

 2.    The methods section would be improved by referring to the temperature reagents were used at, instead of "hot" or "cold" as is the case now.

 Response: The temperature reagents (37°C and 4°C) were added in the experimental section (lines 123, 148 and 173).

 3.    Line 148 states "After this incubation time, unique compact spheroids were formed with reproducibility superior to 98% for both culture conditions." With reference to this sentence what characteristic was reproducible?

 Response: This sentence refers to the reproducibility of spheroid correct formation in each culture plate.

 The following sentence was completed in the manuscript:

 (experimental section, line 151): After this incubation time, unique compact spheroids were formed with reproducibility of spheroid formation per plate superior to 98% for both culture conditions.

 4.    Line 176 (and other places throughout the manuscript) change to "Student's t-test" not "student's t-test".

 Response: The occurrences have been replaced through all the manuscript.

 5.    "Parallely" is used throughout the manuscript, but unfortunately is not a word. "In parallel" would be a better phrase to use. In addition, "more" is used throughout the manuscript, often to begin a sentence, this would be better written as "additionally".

 Response: The occurrences have been changed through all the manuscript.

 6.    The figures appear to have fallen foul of some formatting issues in converting to pdf. There are however some issues that cannot be accounted for in this way. Please remove the boxes and lines from around the graphs. Smoothed lines are used in the line graphs. These are inappropriate in a scientific manuscript, please replace with lines directly between data points. The legends on the graphs, axis labels etc are very difficult to read, please make readable. 

 Response: The quality of figures has been improved, the type of lines in the graph were changed and axis labels have been widened (please see in all figures).  

 7.    Figures 1d, 3a, 3c, 4e, 4f do not have scale bars that are readable, please correct.

 Response: The scale bars were modified in Fig 1d, 3a, 3c, 4e and 4f.

 8.    Figures 4 c, d please remove % from axis numbers, they are not needed.

 Response: The % were removed from figures 4 c, d.

 9.    In the legend for Figure 1 the following is stated: ***** = p<0.00001, however, *** is shown on the graph.

 Response: This sentence was corrected.

 10. Figure 1b shows statistically significant differences in metabolic activity between cells cultured in reference medium and OptiPASS, yet this is not highlighted or discussed.

 The following modifications were made in the manuscript:

 (Results section, line 213-214): It was 0.066±0.007 and 0.052±0.005 at day 1 (p=0.0003), 0.181±0.039 and 0.084±0.006 at day 3 (p=0.0005) and increased to 0.335±0.078 and 0.366±0.056 at day 7 (p=0.40), for reference and OptiPASS medium, respectively (Fig. 1b).

 (Discussion section, lines 486-492): With MDA-MB-231 cell line model, same cell adherence and cell proliferation rates were detected during the experiment in RPMI1640 medium and in OptiPASS medium (at day 7, 78% vs 74%, p=0.64 and 3.0 vs 3.4 p=0.05, respectively). Nevertheless, MDA-MB-231 metabolic activity was firstly lower from Day 1 to Day 3 in OptiPASS medium compared to reference medium. This can be explained by a delayed attachment of the cells in OptiPASS medium that delays the restart of cells metabolic activity. Then, this parameter remained similar from Day 5 until confluence (0.335 vs 0.366 at day 7, p=0.40).

 11. Figure 2d shows low magnification images of SUM1315 cells grown in each medium. At this magnification and level of image reproduction it is difficult to see that there are "round-detached cells in the middle of the wells was detected with OptiPASS medium" as stated in line 230.

 Response: In figure 2d, a magnification of the last well (SUM1315 cells, day 7 in OptiPASS) was added. 

The following modifications were made in the manuscript (Results section, line 253): In contrast, a cluster of round-detached cells in the middle of the wells was detected with OptiPASS medium (Figure 2d, magnification).

 12. In Figures 3 a, c it is not stated whether images are maximum projection from confocal z-stacks or fluorescent images through the center of spheroids. It would aid in understanding if this was clarified.

 Response: The images were captured through the center of the spheroids.

 The following sentence was added in the figure legends, lines 288-289 and 347-348:  Observations were made with Cytation3MV with focal height fixed to the center of each spheroid.

13. Lines 235-236 "More, similar cell repartition profiles were detected by the time (from seeding to confluence), for each culture medium." It is difficult to understand what this statement means. Please clarify the meaning.

 The following sentence was changed in the manuscript:

 (Results section, line 260): However, no difference in cell mortality was detected in both culture conditions (Figure 2c).

 14. In line 269, "level of compaction" is referred to. What is this measurement and how was it estimated/measured?

 Response: This methodology is described in the experimental section of the manuscript and figure S1 (supplementary data).  

 The following sentence was added in the manuscript:

 (Experimental section, lines 152-161): The level of spheroid compaction was determined by the establishment of a correlation between the number of seeded cells and the size of spheroids formed (R2=0.9956) (Figure S1, supplementary data). In this experiment, the spheroids presented similar and high opacity in bright field imaging for all cell seeding concentrations. This methodology was used to estimate the compaction level of spheroids in all experiments.

 15. Lines 310-311 are written in such a way that suggests the drug is having a cytostatic effect on the media, not the cells growing within the media. Again, lines 365-368 should be rewritten from "media" to "cells grown in media"

 The following sentence was changed in the manuscript:

 (Results section, lines 359-360): This highlights the cytostatic activity of epirubicin in these both cell culture conditions.

 and lines 455-457: Similarly, the cellular expression of the mesenchymal marker vimentin remained comparable in cells grown in OptiPASS medium and reference medium, showing the absence of cell line drift with this culture condition.

 16. In lines 315-321, fluorescent signals from live/dead staining are referred to as "markings", "images", "signals", "labelling" would all be more appropriate and scientific.

 Response: All the occurrences of “markings” have been replaced by “labelling”, “staining”, “signals” or “images”.

 17. There was no real explanation as to why Geltrex was used or why it would be preferable to not have animal serum in the medium, but then to have to use an animal derived gel instead.

 Response: Contrary to the serum, which origin cannot be controlled, the Geltrex is produced by EHS controlled cells, extracted and purified by an industrial process. Thus, we tested several batchs of Geltrex (Gibco, ref A1413302: lot 1911983, lot 1965668, lot 2030111, 2052958). All of the tested batchs allowed the formation of reproducible spheroids. Additionally, we tested the development of 3D cell culture with synthetic matrices (Vitrogel, TheWell Bioscience), but only Geltrex permitted the formation of compact spheroids of these two TNBC cell models (Dubois et al., 2017, Oncotarget).

The following sentence was added in the manuscript:

 (Discussion section, lines 445-447): Indeed, contrary to the serum which origin cannot be controlled, the Geltrex® / Matrigel® are produced by EHS controlled cells, extracted and purified by an industrial process [26].

 18. I would like to see some additional argument that it is acceptable to grow SUM1315 in OptiPASS medium. Especially since growing these cells in OptiPASS enhanced proliferation in spheroids, thus changing the model from non-proliferative to proliferative.

 Response: SUM1315 is a highly heterogenous and undifferentiated cell line (claudin-low phenotype) giving it a low proliferation rate in 2D (doubling time > 72h). Thus, the evolution of spheroid size in reference medium was minimal at long term. In contrast, the culture of SUM1315 spheroids in OptiPASS allowed a high increase in spheroid proliferation rate. This is why SUM1315 spheroids grown in reference medium are considered to be non/low-proliferative compared to those grown in OptiPASS medium.

The following sentence was added in the manuscript:

 (Discussion section, lines 507-512): Indeed, SUM1315 is a highly heterogenous and undifferentiated cell line (claudin-low phenotype) giving it a low proliferation rate in 2D (doubling time > 72h). Thus, the evolution of spheroid size in reference medium was minimal at long term. In contrast, the culture of SUM1315 spheroids in OptiPASS allowed a high increase in spheroid proliferation rate. This last property allows the detection of both cytostatic and cytotoxic drug activity, where appropriate.

Reviewer 2 Report

Manuscript ID: jcm-448844 by Dubois et al. describes the new serum-free medium to improve the efficacy of drug screening. It is important to develop the methods to reduce variability of drug screening results and increase reproducibility. The authors introduced new synthetic serum-free OptiPASS medium to achieve it. However, in order to prove the authors’ statements in the manuscript, there are a couple of issues that should be addressed:

 1) Authors showed vimentin immunostaining to prove that cells with OptiPASS maintain mesenchymal function as those with reference media. Vimentin was detected in both culture media, however, cellular localizations are different in each culture condition. In MDA-MB-231, with reference culture medium, vimentin localized more to cytoplasm. But with OptiPASS medium, vimentin is more perinuclear. This indicates that cellular phenotype could already be different in the two culture conditions even though they had the same origin. Similar differences are also observed in SUM1315 cells. It seems to be hard to conclude that two different culture media show similar cell culture performance in 2D with these results.

Furthermore, staining with only one marker is not sufficient. The experiments need to be repeated with additional markers or invasion assay to show maintenance of mesenchymal phenotype.

 2) A major problem of drug screening with cultured cell line is that the same cell line shows different drug responses due to clonal selection during long-term culture in each lab (Ben-David U et al. 2018 Nature). SUM1315 cell lines with OptiPASS in 2D culture showed a lot of floating cells in Figure 2c. This means that clonal selection for adherent cells could occur at 7 days in culture. The selection for the adherent cells could affect drug responses in Figure 4. I suggest that you use very lower passage cells (less than 10 passage, but ideal is less than five passage) and make spheroids with adherent and floating cells. And you can then treat the spheroids with drug to determine if you see a similar result. Furthermore, you also need to confirm if cells are detached from spheroids during the 3D culture. No increase in spheroid size may be due to detachment of the cells from spheroids, instead of cytostatic effect.

 3) In Figure 4e and f, it would have been nice to quantify results with bar graphs that present the number of Ethidium homodimer-1 positive spheroids in total spheroids to show higher cytotoxic effects. It is also unclear what n is (n=5 spheroids? n=5 experiments?).

 Minor comments:

Figure 3d and control experiment in Figure 4b appear to be the same experiment (and thus, the results should look the same, however the spheroid sizes do not match and curve of the lines look different). In figure 3d, spheroid sizes at Day 2 are similar in both media, but those at Day 0 in figure 4b are different. Do you know the reason?

 Figure 4a/4c and 4b/4d, it would be nice if the colors matched. Reference medium is black in Figure 4a and 4b, but OptiPASS is black in Figure 4c and 4d.

 In introduction, line 78 says the non-proliferative SUM1315 spheroids, but they are proliferative in both culture media in Figure 3.

 Author Response

To the Editor:

Dear Ms. Rasic,  

 Thank you very much for your comments and complementary instructions (e-mail received on February the 28th, at the end of the document). Please find attached the revised manuscript untitled:

 The new synthetic serum-free medium OptiPASS promotes high proliferation and drug efficacy prediction on spheroids from MDA-MB-231 and SUM1315 Triple-Negative Breast Cancer cell lines.

By Clémence Dubois, Pierre Daumar, Corinne Aubel, Jean Gauthier, Bernard Vidalinc, Emmanuelle Mounetou, Frédérique Penault-Llorca& Mahchid Bamdad.

 According to your instructions, I applied point-by-point the modifications requested by the reviewers. Please see “Response to reviewer” below.

 Additionally, all the modifications have been done in the manuscript for each Reviewer in the "Track Changes" of Microsoft Word. All the figures have been modified according to reviewers’ instructions and one supplementary data (Figure S1) was added to the manuscript in the experimental section.

 I hope that our manuscript will be published in Journal of Clinical Medicine, I would be at your disposal for any further comment,

 Best regards,

 Mahchid Bamdad

 RESPONSES TO REVIEWER 2

 Open Review

English language and style

( ) Extensive editing of English language and style required
( ) Moderate English changes required
( ) English language and style are fine/minor spell check required
(x) I don't feel qualified to judge about the English language and style

Yes

Can be   improved

Must be   improved

Not   applicable

Does the introduction   provide sufficient background and include all relevant references?

(x)

( )

( )

( )

Is the research design   appropriate?

( )

( )

(x)

( )

Are the methods adequately   described?

(x)

( )

( )

( )

Are the results clearly   presented?

( )

(x)

( )

( )

Are the conclusions   supported by the results?

( )

(x)

( )

( )

Comments and Suggestions for Authors

Manuscript ID: jcm-448844 by Dubois et al. describes the new serum-free medium to improve the efficacy of drug screening. It is important to develop the methods to reduce variability of drug screening results and increase reproducibility. The authors introduced new synthetic serum-free OptiPASS medium to achieve it. However, in order to prove the authors’ statements in the manuscript, there are a couple of issues that should be addressed:

1.     Authors showed vimentin immunostaining to prove that cells with OptiPASS maintain mesenchymal function as those with reference media. Vimentin was detected in both culture media; however, cellular localizations are different in each culture condition. In MDA-MB-231, with reference culture medium, vimentin localized more to cytoplasm. But with OptiPASS medium, vimentin is more perinuclear. This indicates that cellular phenotype could already be different in the two culture conditions even though they had the same origin. Similar differences are also observed in SUM1315 cells. It seems to be hard to conclude that two different culture media show similar cell culture performance in 2D with these results. Furthermore, staining with only one marker is not sufficient. The experiments need to be repeated with additional markers or invasion assay to show maintenance of mesenchymal phenotype.

 Response: As mentioned in the discussion section of the manuscript, the mesenchymal phenotype is mainly characterized by the level of expression of vimentin in various models, including breast cancer (ref 23, 24, 25, 16). This level is furthermore used to quantify the MET transition in cells after treatment (ref 16). In this context, we quantified the vimentin level of expression in our two models cultured in both media. No difference in vimentin expression was detected. More, we input higher resolution images of this staining in Figure 1 and 2 for better demonstration of vimentin cellular localization. These images showed cytoplasmic localization of vimentin for all culture conditions.

            The following sentences were added / modified in the manuscript:

(Experimental section, lines 138-141): The vimentin expression per cell was quantified from the immunofluorescent staining images obtained with both cell lines and culture conditions. For this, we normalized the fluorescent intensity of vimentin on the cellular positive area of each image (Cytation3MV).

 (Results section, line 231-233): Indeed, the vimentin expression in MDA-MB-231 cells was of 8.2±0.2 105 AU and 8.8±0.7 105 AU in reference and OptiPASS medium, respectively (p=0.27).

 (Results section, line 256-257): Indeed, the vimentin expression was of 8.3±0.04 105 and 8.4±0.5 105 AU per cell, respectively (p=0.79).

(Discussion section, lines 417-420): Then the expression of the mesenchymal phenotype marker vimentin was determined. Indeed, vimentin is the most commonly used marker for the determination of a phenotype change from epithelial to mesenchymal for various models, including breast cancers [23],[16],[24],[25].

(Discussion section, lines 433, 435): Similarly, the cellular expression of the mesenchymal marker vimentin remained comparable in cells grown in OptiPASS medium and reference medium, showing the absence of cell line drift with this culture condition.

2.     A major problem of drug screening with cultured cell line is that the same cell line shows different drug responses due to clonal selection during long-term culture in each lab (Ben-David U et al. 2018 Nature). SUM1315 cell lines with OptiPASS in 2D culture showed a lot of floating cells in Figure 2c. This means that clonal selection for adherent cells could occur at 7 days in culture. The selection for the adherent cells could affect drug responses in Figure 4. I suggest that you use very lower passage cells (less than 10 passage, but ideal is less than five passage) and make spheroids with adherent and floating cells. And you can then treat the spheroids with drug to determine if you see a similar result.

 Response: In OptiPASS medium, the cells remain viable whatever the cell repartition i.e. in suspension or attached to the support. In order to limit the clonal selection, the 3D cell culture was formed with all cells including suspended and attached cells cultured in 2D in reference or OptiPASS medium. In these conditions, no difference in spheroid viability (resazurin test) after epirubicin treatment was detected between both culture media (Figure 4a, b). Additionally, the cells were cultured systematically before passage number 30 and that for our purpose of drug screening, no difference in cell repartition profile in 2D has been detected between low passage (n°4) of higher passage (n°30). The suggestion of developing drug screening with only floating cells is very interesting and we will study this in future experiments.

 3.     Furthermore, you also need to confirm if cells are detached from spheroids during the 3D culture. No increase in spheroid size may be due to detachment of the cells from spheroids, instead of cytostatic effect.

Response: No detached cells in bright field imaging were detected during all the experiments. Indeed, the spheroids remained extremely cohesive / compact for the 14 days of culture or the 5 days of treatment, as depicted in live/dead images in Figure 3 and 4, respectively.

4.     In Figure 4e and f, it would have been nice to quantify results with bar graphs that present the number of Ethidium homodimer-1 positive spheroids in total spheroids to show higher cytotoxic effects. It is also unclear what n is (n=5 spheroids? n=5 experiments?).

 Response: The quantification of Ethidium homodimer-1 was realized on epirubicin-treated spheroids by the ratio of Etdh1 intensity in treated spheroids normalized Etdh1 intensity in control spheroids (Cytation3MV). Bar graphs results are represented in Figure 4 g and h for respectively MDA-MB-231 and SUM1315 treated spheroids (mean percentage of Etdh1+ staining with s.d.). 

The following sentences were added in the manuscript:

(Experimental section, lines 185-187) The quantification of Ethidium homodimer-1 was realized on epirubicin-treated spheroids by the ratio of Etdh1 intensity in treated spheroids normalized Etdh1 intensity in control spheroids (Cytation3MV).

(Results section, lines 366-371): The results showed that epirubicin-treated spheroids cultured in reference medium presented overall green staining (with 2±0.5% of ethidium-homodimer positive cells compared to controls, Figure 4f), demonstrating the presence of mainly viable cells composing the spheroids. In contrast, the analysis of treated spheroids cultured in OptiPASS medium showed clearly an increase in dead cells (57±22% of Etdh1+ cells, Figure 4f), revealing also a higher cytotoxic response with this chemotherapeutic agent.

(Results section, lines 332-333): Then, resazurin viability test (Figure 4c, 4d) and Live/Dead fluorescent analysis (Figure 4e, 4d) and ethidium-homodimer-1 intensity quantification in treated-spheroids (Figure 4f, 4g).

(Results section, lines 397-400): Finally, Live/dead images (Figure 4f) and Etdh1+ quantification (4±0.5% and 7±5%, p=0.28, Figure 4h) showed similar viability and mortality patterns between reference and OptiPASS culture conditions, and this for control and epirubicin-treated spheroids.

(Caption, Figure 4, lines 347, 348): (g,h) mean percentage of Etdh1+ staining in treated spheroids compared to control spheroids (error bars = s.d.).

Minor comments:

5.     Figure 3d and control experiment in Figure 4b appear to be the same experiment (and thus, the results should look the same, however the spheroid sizes do not match and curve of the lines look different). In figure 3d, spheroid sizes at Day 2 are similar in both media, but those at Day 0 in figure 4b are different. Do you know the reason?

In introduction, line 78 says the non-proliferative SUM1315 spheroids, but they are proliferative in both culture media in Figure 3.

Response: In the manuscript, we demonstrated that SUM1315 spheroids were more proliferative when cultured in OptiPASS medium compared to reference medium for all experiments. Moreover, (i) SUM1315 is a highly heterogenous and undifferentiated cell line (claudin-low phenotype) (Prat 2010, added to the references, n°31) and (ii) the protocol of spheroid production implies a mix of floating and attached cells that may be explaining the difference in spheroid growth kinetics between experiments. Additionally, the treatment experiment was carried out for five days (Figure 4), compared to a spheroid evolution follow up of 14 days. This can be explaining the difference in spheroid size detected in SUM1315 spheroids in Figure 4.

6.     Figure 4a/4c and 4b/4d, it would be nice if the colors matched. Reference medium is black in Figure 4a and 4b, but OptiPASS is black in Figure 4c and 4d.

 Response: The colors have been modified in the graphs.

Round  2

Reviewer 2 Report

The author's responses seem to be reasonable. I accept their responses.